# Association between Five Common Plasminogen Activator Inhibitor-1 (*PAI-1*) Gene Polymorphisms and Colorectal Cancer Susceptibility

**DOI:** 10.3390/ijms21124334

**Published:** 2020-06-18

**Authors:** Jisu Oh, Hui Jeong An, Jung Oh Kim, Hak Hoon Jun, Woo Ram Kim, Eo Jin Kim, Doyeun Oh, Jong Woo Kim, Nam Keun Kim

**Affiliations:** 1Department of Internal Medicine, CHA Bundang Medical Center, CHA University, Seongnam 13496, Korea; newfascia5@gmail.com (J.O.); doh@cha.ac.kr (D.O.); 2Department of Biomedical Science, College of Life Science, CHA University, Seongnam 13488, Korea; tody2209@naver.com (H.J.A.); jokim8505@gmail.com (J.O.K.); 3Department of Surgery, CHA Bundang Medical Center, CHA University, Seongnam 13496, Korea; iamhacu@chamc.co.kr (H.H.J.); christtome81@chamc.co.kr (W.R.K.); 4Department on Internal Medicine, Asan Medical Center, University of Ulsan College of Medicine, Seoul 05505, Korea; kej7515@hanmail.net

**Keywords:** colorectal cancer, *PAI-1*, polymorphism, susceptibility, miRNA

## Abstract

The plasminogen activator inhibitor-1 (*PAI-1*) is expressed in many cancer cell types and modulates cancer growth, invasion, and angiogenesis. The present study investigated the association between five *PAI-1* gene polymorphisms and colorectal cancer (CRC) risk. Five *PAI-1* polymorphisms (−844G > A [rs2227631], −675 4G > 5G [rs1799889], +43G > A [rs6092], +9785G > A [rs2227694], and +11053T > G [rs7242]) were genotyped using a polymerase chain reaction-restriction fragment length polymorphism assay in 459 CRC cases and 416 controls. Increased CRC risk was more frequently associated with *PAI-1* −675 5G5G polymorphism than with 4G4G (adjusted odds ratio (AOR) = 1.556; 95% confidence interval (CI): 1.012–2.391; *p* = 0.04). In contrast, for the *PAI-1* +11053 polymorphism, we found a lower risk of CRC with the GG genotype (AOR = 0.620; 95% CI: 0.413–0.932; *p* = 0.02) than with the TT genotype, as well as for recessive carriers (TT + TG vs. GG, AOR = 0.662; 95% CI: 0.469–0.933; *p* = 0.02). The +43AA genotype was associated with lower overall survival (OS) than the +43GG genotype. Our results suggest that the *PAI-1* genotype plays a role in CRC risk. This is the first study to identify an association between five *PAI-1* polymorphisms and CRC incidence worldwide.

## 1. Introduction

Colorectal cancer (CRC) is a major cause of cancer morbidity and mortality, accounting for >9% of cancer cases. It is the third most common cancer and the fourth leading cause of cancer-related deaths worldwide [1]. CRC is the second most common cancer in the Republic of Korea, representing approximately 13.6% of all oncological diseases [2]. Available studies have confirmed that the liver is the most common site of metastasis in CRC patients, and metastatic liver cancer is found in 10–25% of patients undergoing surgery due to primary CRC [3]. Local invasiveness and distant metastasis are very important risk factors for CRC. Metastasis and invasion of malignant cancers require proteolytic degradation of the extracellular matrix (ECM) and basement membrane, as well as infiltration of cancer cells into the surrounding tissues, blood stream, or lymphatic vessels. Fibrinolysis is a normal body process. It prevents blood clots that occur naturally from growing and causing problems [4]. Fibrinolysis maintains vessel patency, degrades the ECM, and regulates cell adhesion, migration, and tissue remodeling [4,5,6]. It also regulates cell adhesion, detachment, and migration, playing an important role in cancer progression [5,7,8].

Plasminogen activator inhibitor-1 (*PAI-1*), a 52-kDa glycoprotein belonging to the serine proteinase inhibitor super family, is a multifaceted proteolytic factor located on chromosome 7 (7q21.3–22). It is the principal inhibitor of tissue and urinary plasminogen activators; therefore, it constitutes an important regulatory protein in fibrinolysis [9,10]. Indeed, *PAI-1* is expressed in many cancer cell types and modulates cancer growth, invasion, and angiogenesis in a dose-dependent manner [11]. The regulation of the fibrinolytic system is of critical importance during hemostasis, wound repair, neoplasia, inflammation, and a variety of other biologic processes. This control is achieved in a large part through the action of specific plasminogen activator inhibitors (PAIs). Cultured endothelial cells (ECs) produce type 1 PAI (*PAI-1*), the physiologic inhibitor of tissue-type plasminogen activator. PAI-1 is one of the most highly regulated of the fibrinolytic components produced by ECs. Its synthesis is modulated by a variety of compounds including endotoxin, thrombin, transforming growth factor beta interleukin 1, and tumor necrosis factor alpha [12]. As the functions of *PAI-1* are associated with both tPA and uPA, we endeavored to show unfavorable changes on these vascular risk factors by genetic variations in *PAI-1*.

Increased susceptibility to vascular disease is associated with abnormal *PAI-1* expression [13]. Imbalances in the levels of plasma homocysteine, folate, and urate have been shown to occur in vascular disease [14,15]. Previous investigations have revealed significant correlations between hyperhomocysteinemia and increased *PAI-1* protein levels [16,17]. Future studies are of high significance given the link between unfavorable vascular conditions and *PAI-1* genetic defects.

Recent studies have demonstrated that *PAI-1* is a potent regulator of tumor growth in vivo. Increased *PAI-1* expression has been confirmed in many solid tumor types and is associated with a poor prognosis. Thus, *PAI-1* is considered to be a biochemical marker for poor prognosis in several human cancers and may be a therapeutic target for some cancers [18,19]. There is evidence that plasma *PAI-1* levels correlate closely with rectal cancer metastasis, and tumor tissue *PAI-1* is associated with the histopathology and outcome of rectal cancer [20,21].

Genetic polymorphisms in the *PAI-1* gene appear to contribute to levels of *PAI-1* biosynthesis [22]. According to a previous study [23], the *PAI-1* gene has 10 tag polymorphisms (rs2227631 [−844G > A], rs6092 [+43G > A], rs2227708, rs2227662, rs2227666, rs2227667, rs2227672, rs2227683, rs2227694 [+9785G > A], and rs7242 [+11053T > G]), and rs1799889 [−675 4G > 5G], which has been the focus of numerous *PAI-1* genetic studies. In vitro studies have demonstrated that the *PAI-1* 4G allele produces six times more mRNA than does the 5G allele in response to interleukin-1 [24] and IgE production [25,26,27]. Moreover, an in vivo study detected the highest plasma levels of *PAI-1* in subjects homozygous for the 4G allele and the lowest levels in 5G homozygotes [28]. The polymorphisms 844G > A, +43G > A, +9785G > A, and +11053T > G may also affect plasma *PAI-1* levels [8,18,24,25]. However, the other five polymorphisms (rs2227708, rs2227662, rs2227667, rs2227672, and rs2227683) do not influence *PAI-1* protein levels. Therefore, we designed a genetic epidemiological study of five *PAI-1* polymorphisms (*PAI-1* −675 4G > 5G [rs1799889], −844G > A [rs2227631], +43G > A [rs6092], +9785G > A [rs2227694], and +11053T > G [rs7242]) to investigate the association between *PAI-1* and CRC. This is the first study to identify an association between five *PAI-1* polymorphisms and CRC incidence worldwide.

## 2. Results

### 2.1. Baseline Characteristics of Colorectal Cancer and Controls

Table 1 shows the main characteristics of the study participants. The mean age of the patients was 61.5 ± 12.6 years and that of the controls was 61.0 ± 11.4 years. Cases had more frequent HTN (hypertension) and DM (diabetes mellitus) than controls. In addition to BMI, levels of folate, TG (triglyceride), and HDL-C (high-density lipoprotein-cholesterol) were lower in cases than in controls. Of the 459 cases, 51% were diagnosed with advanced stage disease (AJCC 7th cancer stage III or IV).

### 2.2. PAI-1 -675 and PAI-1 +11053 Gene Polymorphisms are Associated with Colorectal Cancer Risk

Five *PAI-1* gene polymorphisms were amplified. The genotype and allele frequencies were in Hardy–Weinberg equilibrium (*p* > 0.05) in the control and patients’ individuals for all polymorphisms (Table 2). Comparison of the genotype frequencies and adjusted odds ratio (AOR) values of *PAI-1* gene polymorphisms between the CRC and control subjects is presented in Table 2 and Appendix A. We found a lower risk of PAI-1 +11053T > G with the GG genotype (AOR = 0.594; 95% CI: 0.370–0.954; *p* = 0.03) than with the TT genotype, as well as for recessive carriers (TT + TG vs. GG, AOR = 0.667; 95% CI: 0.446-0.990; *p* = 0.05) (Appendix A). Because CRC has been shown to be influenced by various environmental factors, we performed a stratified analysis of age, gender, HTN, DM, and levels of peripheral blood factors (homocysteine, folate, TG, and HDL) to determine whether any associations between *PAI-1* polymorphisms and CRC risk exist.

These results were adjusted for age, gender, HTN, and DM. A higher risk of CRC was found for homozygous carriers of the *PAI-1* -675 5G5G polymorphism than with 4G4G [AOR = 1.556; 95% confidence interval (CI): 1.012–2.391; *p* = 0.04]. In contrast, for the *PAI-1* +11053 polymorphism, we found a lower risk of CRC in the GG genotype (AOR = 0.620; 95% CI: 0.413–0.932; *p* = 0.02) than in the TT genotype, or in recessive carriers (TT + TG vs. GG, AOR = 0.662; 95% CI: 0.469–0.933; *p* = 0.02). We found no major differences in the association of other variants with CRC.

### 2.3. PAI-1 Gene Polymorphism and Patient Clinicopathological Factors Affect CRC Susceptibility

The associations between these polymorphisms and patient clinicopathological factors are shown in Table 3 and Appendix A. *PAI*-I -675 4G5G + 5G5G genotype showed an increased risk of CRC for patients aged < 61 years (AOR = 1.942; 95% CI: 1.247–3.024; *p* = 0.01), without HTN (AOR = 1.694; 95% CI: 1.093–2.627; *p* = 0.02), without DM (AOR = 1.464; 95% CI: 1.051–2.039; *p* = 0.02), with folate ≥ 3.8 ng/mL (AOR = 1.449; 95% CI: 1.030–2.039; *p* = 0.03), and with homocysteine ≤ 13.2 µmol/L (AOR = 1.630; 95% CI: 1.163–2.285; *p* = 0.01). However, the *PAI-1* +11053TG + GG genotype was related to a protective effect against CRC for patients aged < 61 years (AOR = 0.563; 95% CI: 0.333–0.950; *p* = 0.03), with HTN (AOR = 0.618; 95% CI: 0.382–1.001; *p* = 0.05), without DM (AOR = 0.633; 95% CI: 0.425–0.942; *p* = 0.02), with folate ≥ 3.8 ng/mL (AOR = 0.669; 95% CI: 0.448–0.999; *p* = 0.05), and with BMI ≥ 25 (AOR = 0.420; 95% CI: 0.192–0.920; *p* = 0.03). However, after considering multiple comparisons, we concluded that the association of 4G5G + 5G5G, the association of 11053TG + GG, and the association of 9785GA polymorphism had no relation with CRC. The *PAI-1* -675 4G5G + 5G5G genotype with HTN was three times more likely to develop CRC than the 4G4G genotype without HTN and was five times more likely to develop CRC in patients with a folate level ≤ 3.8 nmol/L (Figure 1 and Appendix A). Moreover, the *PAI-1* 43GA + AA genotype with DM were three times more likely to increase the CRC incidence than the GG genotype without DM (Figure 1 and Appendix A).

### 2.4. The Five Combined PAI-1 Polymorphisms Have Different Effects on CRC Occurrence

We conducted haplotype analysis to evaluate the combined effects of five *PAI-1* polymorphisms on CRC occurrence (Table 4). The *PAI-1* -844G/-675 4G/+43A/+9785G/+11053G haplotype (AOR = 0.045; 95% CI: 0.003–0.780; *p* = 0.01; false-positive discovery rate (FDR)-corrected *p* = 0.01) and -844A/-675 4G/+43A/+9785G/+11053G haplotype (AOR = 0.121; 95% CI: 0.041–0.360; *p* < 0.01; FDR-corrected *p* = 0.01) were associated with a greatly decreased risk of CRC. In contrast, the -844A/-675 5G/+43G/+9785G/+11053T haplotype had an increased risk (AOR = 3.733; 95% CI: 1.354–10.290; *p* = 0.01; FDR-corrected *p* = 0.05). Other combined haplotypes did not affect the incidence of CRC, or the differences were not statistically significant.

### 2.5. The PAI-1 +43AA Genotype is Associated with Poor Survival

The associations between *PAI-1* polymorphisms and CRC patient survival are shown in Figure 2. Multivariate Cox proportional analysis (Appendix A) showed that only the *PAI-1* +43 genotype had a significant effect on survival. Compared with the +43GG genotype, the +43AA genotype was associated with poor OS (overall survival) (HR = 8.551; 95% CI: 1.833–39.89; *p* = 0.01; Figure 2A) and relapse-free survival (RFS) (HR = 12.71; 95% CI: 1.330–121.3; *p* = 0.03; Figure 2C). Moreover, OS (HR = 9.330; 95% CI: 2.043–42.61; *p* = 0.01; Figure 2B) and RFS (HR = 11.97; 95% CI: 1.433–100.0; *p* = 0.02; Figure 2D) were associated with the recessive genotype (GG + GA vs. AA). However, no associations between other polymorphisms and survival outcomes were found.

## 3. Discussion

In this study, we investigated the effect of five *PAI-1* gene polymorphisms (-675 4G > 5G [rs1799889], -844G > A [rs2227631], +43G > A [rs6092], +9785G > A [rs2227694], and +11053T > G [rs7242]) on the risk of CRC in a Korean population-based case-control study. We demonstrated that the *PAI-1* -675 4G > 5G and +11053T > G polymorphisms were associated with increased CRC incidence. Patients with the 5G5G genotype had a significantly higher risk of CRC than those with the 4G5G and 4G4G genotypes of the *PAI-1* -675 polymorphism. A reduced risk was found for the *PAI-1* +11053GG genotype.

In addition, survival among the *PAI-1* +43AA genotype was lower than in the GG genotype. To date, most studies of *PAI-1* gene polymorphisms have focused on -675 4G > 5G and -844G > A. However, studies of the other three polymorphisms are negligible. These findings suggest that +43G > A and +11053T > G are meaningful. This is the first study to investigate the association between five *PAI-1* polymorphisms and CRC susceptibility and prognosis.

The other part of our analysis focused on the −675 4G > 5G polymorphisms of *PAI-1* in patients with CRC. Previous studies have shown that the −675 4G > 5G promoter polymorphism of *PAI-1* may modulate *PAI-1* transcription. The 4G allele insertion/deletion promoter polymorphism has been associated with elevated plasma levels of PAI-1 [29,30]. The *PAI-1* 4G allele does seem to play a role in CRC progression according to several studies [31,32,33,34]. A study by Loktionov et al. [35] found an insignificant reduction in developing CRC compared with individuals with the 5G5G genotype among 206 patients with CRC and 355 controls for *PAI-1* 4G5G heterozygotes (OR = 0.82; 95% CI: 0.49–1.35) and no risk reduction for 4G4G carriers (OR = 1.05; 95% CI: 0.60–1.82) compared with individuals with the 5G5G genotype. However, another study by Försti et al. including 308 patients and 585 controls found no reduced risk of CRC for 4G5G and 4G4G carriers. Additionally, the results of Vossen et al. [36] did not support an effect of *PAI-1* 4G5G on the CRC risk. Therefore, the *PAI-1* 4G allele might not have a clear role in the onset of CRC. The variable and contradictory results of other studies [31,32,33,34,35,36] may be evidence that interactions with other factors, such as environmental factors as well as genes, have an influence in the development of cancer. However, further specifically designed studies are needed to assess their value in this respect.

PAI-1 has been shown to be crucial in the plasminogen activator-plasmin protease system. It plays an important role in a wide variety of physiologic and pathologic processes, including fibrosis, fibrinolysis, wound healing, and cancer metastasis. Many reports indicate that several common *PAI-1* gene polymorphisms are risk factors for various diseases related to the elevated serum levels of PAI-1. The mechanism of how polymorphisms of PAI-1 gene affect the progression and metastasis of cancer may directly result from opposing effects of the two alleles at the transcription level [37], a finding that should be regarded in light of PAI-1 interactions with other components of the uPA system. In addition to modulating extracellular matrix degradation [38,39], *PAI-1* was recently shown to be a potent regulator of cancer-related angiogenesis [40,41,42]. Experiments in *PAI-1* deficient mice have demonstrated impaired angiogenic responses in these animals, contributing to the prevention of invasion and growth of transplanted cancer cells [43,44]. Additional direct mechanisms of tumor growth and invasion stimulation by PAI-1 may include enhancement of cancer cell migration [45] and inhibition of apoptosis [46].

We also found that HTN or DM increased the risk of CRC, independent of genotype (Table 3 and Appendix A). The CRC susceptibility differs depending on clinicopathologic factors such as age, presence of HTN, existence of DM, obesity, folate level, and homocysteine level (Figure 1). The *PAI-1*-675 4G5G + 5G5G genotype without HTN or DM was associated with a higher risk of CRC than the 4G4G genotype, but the *PAI*-1 +11053TT + TG genotype with HTN or without DM had a lower risk of CRC than did the GG genotype. Moreover, at folate levels ≥3.8 ng/mL, the *PAI-1* -675 4G5G + 5G5G genotype positively correlated with CRC risk, but the *PAI-1* 11053GG genotype was inversely correlated. These findings suggest that genetic and environmental factors, such as metabolic diseases, influence each other in CRC and may implicate *PAI-1* polymorphisms as pathophysiologic linkers/modulators between metabolic syndrome and cancer [13,14,15,16,17].

The rs7242 polymorphism is located in the 3′-untranslated region (UTR) of PAI-1 and is characterized by the substitution of a guanine with thymine. Our association study of PAI-1 3′-UTR polymorphism (+11053T > G) is the first report of CRC. Marchand et al. reported that miR-421 and miR-30c inhibit *PAI-1* expression by binding to *PAI-1* 3′-UTR [47]. Recent reports have suggested that miR-30a and miR-30b effectively inhibited plasminogen activator inhibitor-1 (PAI-1) in NSCLC cells [48]. MiR-30b expression was decreased in pancreatic cancer patients with diabetes. MiR-30b could bind with the 3′-UTR seed region of *PAI-1* mRNA to regulate its expression [49]. MiRNA is predicted in the pathological process induced by CRC by regulating *PAI-1*. Finally, further research is needed to determine whether the polymorphism of rs7242 affects the expression of *PAI-1* by these miRNAs.

Our study has several limitations. First, well-known, extensively studied *PAI-1* polymorphisms were obtained from a well-defined homogeneous population of a single ethnic group. Second, only limited data on these *PAI-1* polymorphisms are presented, although the polymorphisms appear functional. Third, the manner in which these polymorphisms affect CRC development remains unclear. Fourth, the lack of information on coagulation activity and *PAI-1* and inflammatory cytokine expression in CRC patients remains to be investigated. Finally, the study population was restricted to Koreans; other ethnic populations should be studied in the future.

If FDR-adjusted *p*-values had been used as criteria for statistical significance, the results would have been more conclusive. However, it might have been difficult to obtain such results in our study because we divided our patient population into many groups for analysis of all five *PAI-1* polymorphisms. Moreover, it is important that the AORs are statistically significant.

In conclusion, our results support the relationship between CRC susceptibility and *PAI-1* gene polymorphisms. Our study suggests that the -675 5G5G genotype is associated with increased CRC susceptibility, but vice versa for the +11053GG genotype. Additionally, we found evidence that the +43AA genotype may be related to an unfavorable CRC prognosis, and CRC development is affected by genetic and environmental factors, such as metabolic diseases. Thus, further studies with larger and more heterogeneous cohorts are needed to extend our understanding of the influence of polymorphisms in *PAI-1* genes on CRC. If additional research identifies a definitive causative role for the *PAI-1* pathway in CRC pathogenesis, regulation of *PAI-1* expression or activity may provide preventive options for CRC. Further research into the association between fibrinolysis and cancer could lead to the discovery of novel drug targets for cancer prevention or treatment.

## 4. Materials and Methods

### 4.1. Population and Clinical Samples

This study protocol was reviewed and approved by the Institutional Review Board of CHA Bundang Medical Center. All study subjects provided written informed consent to participate in the study. All methods were conducted in accordance with approved guidelines and regulations. Blood samples were collected from 459 Korean patients who were diagnosed with CRC at the CHA Bundang Medical Center of CHA University (Seongnam, Korea) between June 2004 and January 2009. The Institutional Review Board of CHA Bundang Medical Center approved this genetic study in June 2009 (IRB No. PBC09-077) and informed consent was obtained from study participants.

We retrospectively obtained patient information concerning age, gender, underlying conditions of hypertension (HTN), diabetes mellitus (DM), and body mass index (BMI), tumor size, stage, and site, time to progression, and time to mortality. We estimated the levels of homocysteine, triglyceride (TG), high-density lipoprotein cholesterol (HDL-C), and folate. The American Joint Committee on Cancer (AJCC)’s *Classification and Stage Groupings*, 7th edition, was used for tumor assessment. The cancer-free control group consisted of 416 individuals who were randomly selected from participants in a health-screening program and excluded those with a history of cancer and other medical diseases. All study subjects were ethnic Koreans.

### 4.2. Polymorphism Analysis

DNA was extracted from leukocytes using a G-DEX II Genomic DNA Extraction Kit (Intron Biotechnology, Seongnam, Republic of Korea) [50], according to the manufacturer’s instructions. To analyze *PAI-1* genotypes, we used polymerase chain reaction-restriction fragment length polymorphism (PCR-RFLP) analysis because this procedure is more cost-effective than the sequencing of entire genes.

The *PAI-1* polymorphism −844G > A (rs2227631) was detected using forward (5′-CAG GCT CCC ACT GAT TCT AC-3′) and reverse (5′-GAG GGC TCT CTT GTG TCA AC-3′) primers. The 510-bp PCR product was then digested with *Xho*I. A digestion product of 510 bp represented the AA genotype; fragments of 510 bp, 364 bp, and 146 bp represented the GA genotype; and the 364-bp and 146-bp products represented the GG genotype.

The *PAI-1* –675 4G > 5G (rs1799889) polymorphism was detected by PCR-RFLP analysis using forward (5′-CCA ACA GAG GAC TCT TGG TC-3′) and reverse (5′-CAC AGA GAG AGT CTG GCC ACG-3′) primers. The 99-bp product was digested with 3 U *Bsl*I for 16 h at 55 °C. A restriction fragment of 99 bp represented the 4G4G genotype; fragments of 99 bp, 77 bp, and 22 bp represented the 4G5G genotype; and the 77-bp and 22-bp products represented the 5G5G genotype.

To detect the *PAI-1* +43G > A (rs6092) genotype, PCR-RFLP analysis was performed with forward (5′-TGT CTT CCA GAA CGA TTC CTT CAC C-3′) and reverse (5′-GTT GTC AGC TGG AGC ATG GCC-3′) primers. The amplified fragment was 266 bp in length. The PCR products were digested with 3 U *Psh*AI for 16 h at 37 °C. A restriction product of 266 bp identified the GG genotype; products of 266 bp, 172 bp, and 94 bp represented the GA genotype; and the 172-bp and 94-bp products represented the AA genotype.

The *PAI-1* +9785G > A (rs2227694) genotype was amplified using forward (5′-ATG AAG GTG CCA CTG CAC TCG C-3′) and reverse (5′-ATG ACA GCT GCA AGA GAA GGG ACA-3′) primers. The length of the amplified fragment was 135 bp. PCR products were digested with 3 U *Hha*I for 20 h at 37 °C. Restriction fragments of 112 bp and 23 bp represented the GG phenotype; fragments of 135 bp, 112 bp, and 23 bp represented the GA genotype; and a fragment of 135 bp represented the AA genotype.

*PAI-1* +11053T > G (rs7242) was amplified using forward (5′-CCT CCC CAG AAA CAG TGT GCA TGG G-3′) and reverse (5′-AAG AGC TGG GCA CGC ATC TGG C-3′) primers. The amplified fragment was 130 bp in length. PCR products were digested with 3 U *Hae*III for 16 h at 37 °C. A restriction fragment of 130 bp represented the TT genotype; fragments of 130 bp, 109 bp, and 21 bp represented the TG genotype; and fragments of 109 bp and 21 bp represented the GG genotype.

The accuracy of these genotyping methods was confirmed by sequencing 30% of randomly selected samples, followed by DNA sequencing, to validate the RFLP findings. Sequencing was performed using an ABI 3730xl DNA Analyzer (Applied Biosystems, Foster City, CA, USA). The concordance of the quality control samples was 100%.

### 4.3. Statistical Analysis

The observed genotype frequencies in the controls were tested for Hardy–Weinberg equilibria, and the difference between the observed and expected frequencies was tested for significance using the Fisher’s exact test. Statistical significance of the differences in the genotype frequencies between the CRC cases and controls was determined using the Chi-squared test. Odds ratios (ORs) and 95% confidence intervals (CIs) for associations between the genotypes and CRC were calculated using GraphPad Prism 4.0 (GraphPad Software, Inc., San Diego, CA, USA) and MedCalc, version 12.1.4 (MedCalc Software, Mariakerke, Belgium). A combination analysis of the *PAI-1* genotypes was performed as described previously; the combined genotypes were used to avoid rare frequencies, diminishing their statistical power. *PAI-1* haplotypes were estimated using SNPAlyze, version 5.1 (DYNACOM Co. Ltd., Yokohama, Japan), and the estimated *PAI-1* haplotype frequencies were compared using the Fisher’s exact test. Overall survival (OS) was assessed from the date of diagnosis to the date of death from any cause or the date of the last visit; relapse-free survival (RFS) was defined as the time of diagnosis to the first evidence of clinical or radiographic relapse or the date of the last follow-up. Survival was compared using the Kaplan–Meier method, and *PAI-1* gene polymorphisms were verified by multivariate analysis using a Cox regression model. Hazard ratios were calculated using a 95% CI. All tests were two-tailed and a *p*-value less than 0.05 was considered statistically significant.

## Figures and Tables

**Figure 1 ijms-21-04334-f001:**
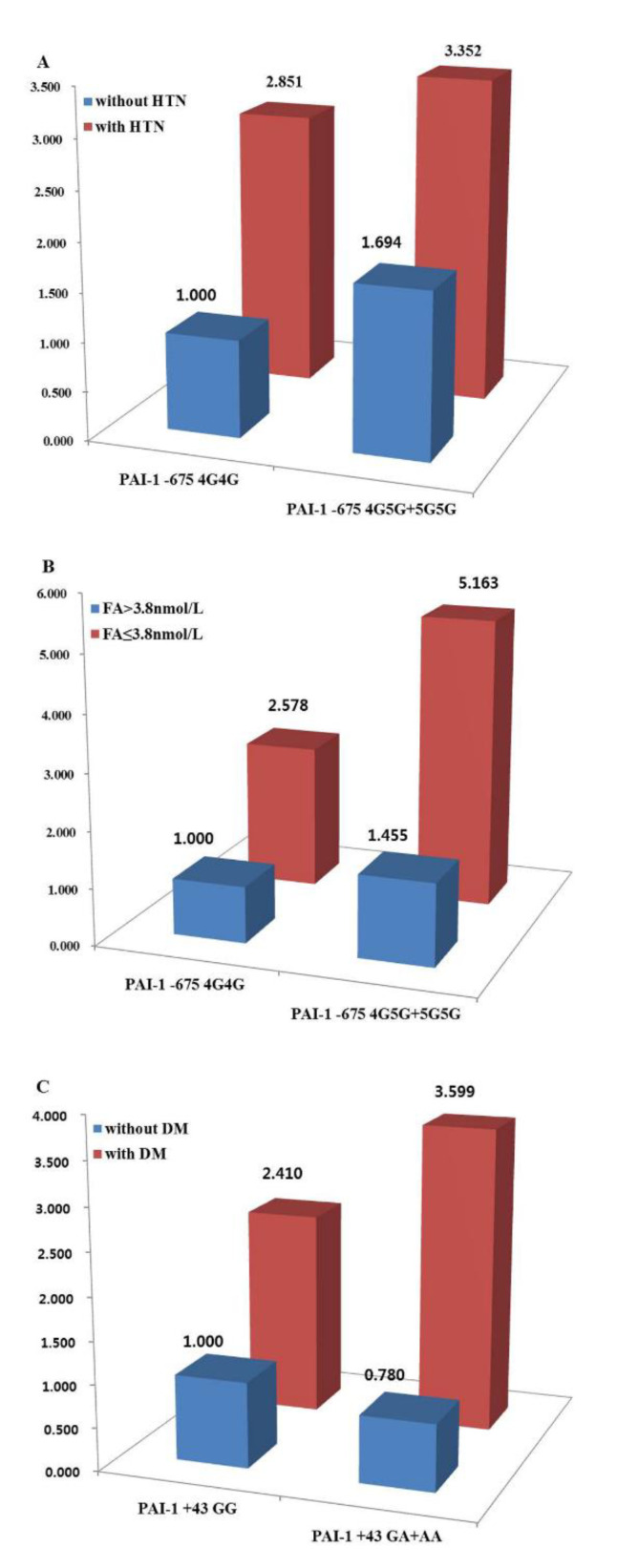
Colorectal cancer incidence (odds ratio) by interactions between genes and environmental factors such as (**A**) hypertension (HTN), (**B**) folate (FA), and (**C**) diabetes mellitus (DM).

**Figure 2 ijms-21-04334-f002:**
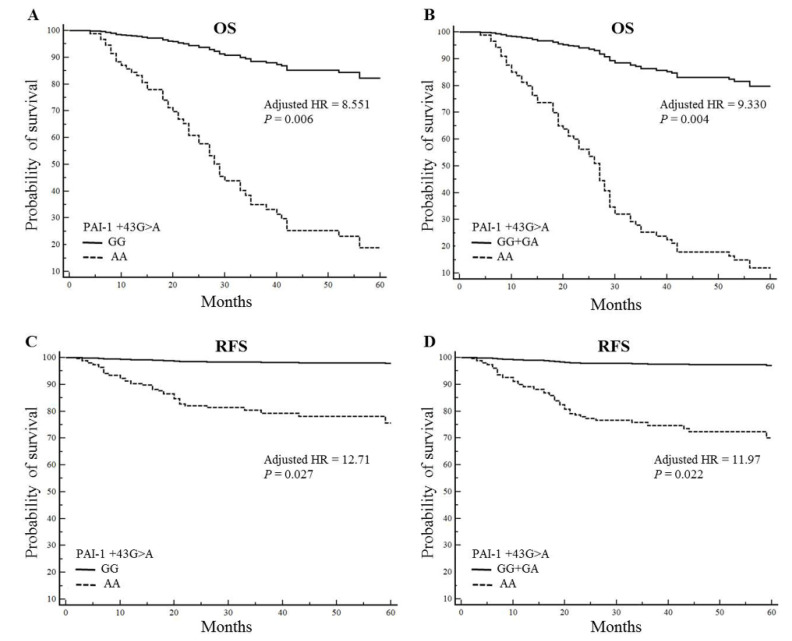
Overall survival (OS; A, B) rates and relapse-free survival (RFS; C, D) rate of the *PAI-1* +43G > A polymorphism in colorectal cancer patients. (**A**) OS curves of patients in the *PAI-1* +43GG vs. +43AA group. (**B**) OS curves of patients in the *PAI-1* +43GG + GA vs. +43AA group. (**C**) RFS curves of patients in the *PAI-1* +43GG vs. +43AA group. (**D**) RFS curves of patients in the *PAI-1* +43GG + GA vs. +43AA group.

**Table 1 ijms-21-04334-t001:** Baseline characteristics of colorectal cancer patients and control subjects.

Characteristic	Control	Case	*p*	Colon	*p*	Rectum	*p*
N	416	459		268		191	
Age (mean ± SD)	61.04 ± 11.44	61.52 ± 12.63	0.56	61.00 ± 12.96	0.966	61.79 ± 12.12	0.47
Gender (male), n (%)	173 (41.6)	215 (48.0)	0.25	124 (47.1)	0.394	91 (49.2)	0.30
Anti-HTN drug or BP ≥ 140/90 mmHg, n (%)	186 (44.7)	284 (63.4)	0.01	163 (62.0)	0.016	121 (65.4)	0.01
Anti-DM drug or FBS ≥ 126 mg/dl, n (%)	61 (14.7)	159 (35.5)	<0.01	91 (34.6)	<0.0001	68 (36.8)	<0.01
Homocysteine (µmol/L)	9.78 ± 4.22	10.47 ± 7.85	0.13	10.25 ± 8.20	0.349	10.70 ± 7.40	0.07
Folate (ng/mL)	9.10 ± 8.05	7.82 ± 6.35	0.02	7.82 ± 5.97	0.039	7.80 ± 6.93	0.08
Triglycerides (mg/dl)	143.78 ± 87.50	126.41 ± 77.42	0.01	126.65 ± 81.04	0.016	125.70 ± 73.21	0.02
Body mass index (kg/m^2^)	24.26 ± 3.36	23.09 ± 3.17	<0.01	22.91 ± 3.28	<0.0001	23.28 ± 2.99	0.01
Tumor size							
<5 cm	-	189 (42.2)	-	101 (38.4)	-	88 (47.6)	-
≥5 cm	-	270 (60.3)	-	167 (63.5)	-	103 (55.7)	-
TNM stage, n (%)							
I	-	45 (10.0)	-	23 (8.7)	-	22 (11.9)	-
II	-	183 (40.8)	-	113 (43.3)	-	69 (37.3)	-
III	-	183 (40.8)	-	102 (38.8)	-	81 (43.8)	-
IV	-	48 (10.7)	-	29 (11.0)	-	19 (10.3)	-

Standard deviation (SD), blood pressure (BP), fasting blood sugar (FBS), hypertension (HTN), diabetes mellitus (DM), high-density lipoprotein-cholesterol (HDL-C), tumor node metastasis (TNM). The *p* values were calculated by Student’s t-test for continuous variables and Chi-squared test for categorical variables.

**Table 2 ijms-21-04334-t002:** Comparison of the genotype frequencies and AOR (adjusted odds ratio) values of *PAI-1* gene polymorphisms between the colorectal cancer and control subjects.

Genotype	Controls (*n* = 416)	CRC Patients (*n* = 459)	AOR (95% CI) *	*p* ^a^	*p* ^b^
*PAI-1* -844G > A					
GG	136 (32.7)	154 (33.6)	1.000 (reference)		
GA	199 (47.8)	230 (50.1)	1.000 (0.730–1.370)	1.00	1.00
AA	81 (19.5)	75 (16.3)	0.779 (0.508–1.195)	0.25	0.42
Dominant (GG vs. GA + AA)			0.937 (0.695–1.262)	0.67	0.83
Recessive (GG + GA vs. AA)			0.806 (0.559–1.164)	0.25	0.42
HWE-*P*	0.592	0.415			
*PAI-1*-675 4G > 5G					
4G4G	180 (43.3)	171 (37.3)	1.000 (reference)		
4G5G	180 (43.3)	206 (44.9)	1.212 (0.890–1.651)	0.22	1.00
5G5G	56 (13.5)	82 (17.9)	1.556 (1.012–2.391)	0.04	0.11
Dominant (4G4G vs. 4G5G + 5G5G)			1.284 (0.963–1.714)	0.09	0.45
Recessive (4G4G + 4G5G vs. 5G5G)			1.385 (0.938–2.044)	0.10	0.26
HWE-*P*	0.306	0.128			
*PAI-1* +43G > A					
GG	335 (80.5)	375 (81.7)	1.000 (reference)		
GA	75 (18.0)	70 (15.3)	0.890 (0.612–1.296)	0.54	1.00
AA	6 (1.4)	14 (3.1)	0.647 (0.155–2.694)	0.55	0.69
Dominant (GG vs. GA + AA)			0.875 (0.606–1.261)	0.47	0.79
Recessive (GG + GA vs. AA)			0.670 (0.161–2.779)	0.58	0.73
HWE-*P*	0.447	0.851			
*PAI-1* +9785G > A					
GG	383 (92.1)	417 (90.8)	1.000 (reference)		
GA	31 (7.5)	42 (9.2)	1.079 (0.629–1.849)	0.78	1.00
AA	2 (0.5)	0 (0.0)	N/A	1.00	1.00
Dominant (GG vs. GA + AA)			1.000 (0.588–1.700)	1.00	1.00
Recessive (GG + GA vs. AA)			N/A	1.00	1.00
HWE-*P*	0.124	0.329			
*PAI-1* +11053T > G					
TT	107 (25.7)	133 (29.0)	1.000 (reference)		
TG	204 (49.0)	241 (52.5)	0.966 (0.692–1.349)	0.84	1.00
GG	105 (25.2)	85 (18.5)	0.620 (0.413–0.932)	**0.02**	0.11
Dominant (TT vs. TG + GG)			0.850 (0.620–1.165)	0.31	0.78
Recessive (TT + TG vs. GG)			0.662 (0.469–0.933)	**0.02**	0.10
HWE-*P*	0.695	0.200			

* The adjusted odds ratio on the basis of risk factors, such as age, gender, hypertension, and diabetes mellitus. ^a^
*p*-value calculated by multiple logistics regression analysis. ^b^ False-positive discovery rate (FDR)-adjusted *p*-value.

**Table 3 ijms-21-04334-t003:** Stratified effects of PAI-1 gene polymorphisms on colorectal cancer.

Variable	*PAI-1* -844GA			*PAI-1* -675 4G5G + 5G5G			*PAI-1* +43GA + AA			*PAI-1* 9785GA			*PAI-1* 11053TT + TG		
	AOR (95% CI)	*p ^a^*	*p ^b^*	AOR (95% CI)	*p ^a^*	*p ^b^*	AOR (95% CI)	*p ^a^*	*p ^b^*	AOR (95% CI)	*p ^a^*	*p ^b^*	AOR (95% CI)	*p ^a^*	*p ^b^*
Age															
<61 years	1.273 (0.801–2.023)	0.31	0.38	1.942 (1.247–3.024)	0.01	0.02	0.889 (0.528–1.495)	0.66	0.66	0.633 (0.297–1.349)	0.24	0.38	0.563 (0.333–0.950)	0.03	0.08
≥61 years	0.814 (0.517–1.280)	0.37	0.62	1.034 (0.692–1.544)	0.87	0.87	0.872 (0.509–1.494)	0.62	0.77	2.467 (1.028–5.920)	0.04	0.22	0.741 (0.462–1.188)	0.21	0.53
Gender															
Male	0.881 (0.536–1.447)	0.62	0.77	1.313 (0.824–2.093)	0.25	0.74	1.029 (0.568–1.862)	0.93	0.93	1.306 (0.483–3.532)	0.60	0.77	0.752 (0.440–1.285)	0.30	0.74
Female	1.141 (0.745–1.748)	0.55	0.68	1.437 (0.980–2.106)	0.06	0.16	0.765 (0.472–1.241)	0.28	0.46	1.123 (0.572–2.206)	0.74	0.74	0.600 (0.378–0.954)	0.03	0.16
HTN															
without	1.367 (0.852–2.194)	0.20	0.33	1.694 (1.093–2.627)	0.02	0.10	0.980 (0.575–1.669)	0.94	0.94	1.395 (0.670–2.907)	0.37	0.47	0.668 (0.398–1.120)	0.13	0.32
with	0.793 (0.501–1.255)	0.32	0.67	1.138 (0.755–1.715)	0.54	0.67	0.838 (0.493–1.424)	0.51	0.67	0.942 (0.415–2.139)	0.89	0.89	0.618 (0.382–1.001)	0.05	0.25
DM															
without	1.236 (0.861–1.776)	0.25	0.31	1.464 (1.051–2.039)	0.02	0.06	0.780 (0.516–1.179)	0.24	0.31	1.268 (0.693–2.320)	0.44	0.44	0.633 (0.425–0.942)	0.02	0.06
with	0.410 (0.180–0.937)	0.03	0.17	1.017 (0.521–1.986)	0.96	0.96	1.445 (0.525–3.978)	0.48	0.96	0.829 (0.223–3.076)	0.78	0.96	0.820 (0.383–1.758)	0.61	0.96
Smoking															
without	1.070 (0.732–1.564)	0.73	0.73	1.351 (0.962–1.898)	0.08	0.21	0.798 (0.516–1.234)	0.31	0.41	1.405 (0.710–2.780)	0.33	0.41	0.679 (0.455–1.014)	0.06	0.21
with	0.839 (0.454–1.552)	0.58	0.74	1.527 (0.837–2.787)	0.17	0.47	1.130 (0.554–2.307)	0.74	0.74	0.771 (0.270–2.202)	0.63	0.74	0.616 (0.301–1.263)	0.19	0.47
* Folate															
≥3.8 ng/mL	1.065 (0.733–1.546)	0.74	0.74	1.449 (1.030–2.039)	0.03	0.12	0.878 (0.572–1.347)	0.55	0.74	1.164 (0.623–2.175)	0.63	0.74	0.669 (0.448–0.999)	0.05	0.12
<3.8 ng/mL	0.669 (0.256–1.750)	0.41	0.79	1.783 (0.760–4.186)	0.18	0.79	0.652 (0.202–2.103)	0.47	0.79	N/A	0.99	0.99	1.183 (0.448–3.121)	0.74	0.92
** Homocysteine															
<13.2 µmol/L	0.911 (0.633–1.312)	0.62	0.63	1.630 (1.163–2.285)	0.01	0.03	0.743 (0.480–1.151)	0.18	0.31	1.168 (0.626–2.179)	0.63	0.63	0.681 (0.461–1.005)	0.05	0.13
≥13.2 µmol/L	1.875 (0.739–4.757)	0.19	0.79	0.895 (0.390–2.052)	0.79	0.79	1.275 (0.450–3.616)	0.65	0.79	0.644 (0.080–5.201)	0.68	0.79	1.200 (0.455–3.163)	0.71	0.79
BMI															
<25 kg/m^2^	0.832 (0.529–1.308)	0.43	0.65	1.413 (0.935–2.135)	0.10	0.25	1.044 (0.615–1.772)	0.87	0.87	1.318 (0.565–3.071)	0.52	0.65	0.672 (0.420–1.075)	0.10	0.25
≥25 kg/m^2^	1.061 (0.546–2.063)	0.86	0.86	1.333 (0.715–2.488)	0.37	0.70	1.211 (0.523–2.803)	0.66	0.82	1.596 (0.513–4.963)	0.42	0.70	0.420 (0.192–0.920)	0.03	0.15
* VB_12_															
≥368 mg	0.843 (0.304–2.339)	0.74	0.74	0.724 (0.312–1.679)	0.45	0.74	0.436 (0.098–1.944)	0.28	0.74	0.643 (0.079–5.260)	0.68	0.74	0.562 (0.184–1.711)	0.31	0.74
<368 mg	N/A	N/A	N/A	N/A	N/A	N/A	N/A	N/A	N/A	N/A	N/A	N/A	N/A	N/A	N/A
Triglycerides															
<126.65 mg/dL	1.039 (0.648–1.664)	0.88	0.88	1.532 (1.010–2.323)	0.05	0.23	0.866 (0.512–1.465)	0.59	0.74	0.599 (0.237–1.513)	0.28	0.46	0.657 (0.401–1.078)	0.10	0.24
≥126.65 mg/dL	1.087 (0.644–1.836)	0.76	0.92	1.025 (0.628–1.672)	0.92	0.92	0.807 (0.424–1.536)	0.51	0.92	1.172 (0.516–2.659)	0.71	0.92	0.697 (0.396–1.224)	0.21	0.92
Cholesterol															
<150 mg/dL	0.747 (0.265–2.107)	0.58	0.95	1.083 (0.417–2.812)	0.87	0.95	0.618 (0.165–2.318)	0.48	0.95	1.048 (0.243–4.524)	0.95	0.95	1.316 (0.471–3.675)	0.60	0.95
≥150 mg/dL	1.157 (0.722–1.854)	0.55	0.68	1.254 (0.814–1.931)	0.31	0.51	1.088 (0.638–1.855)	0.76	0.76	1.612 (0.761–3.413)	0.21	0.51	0.560 (0.328–0.958)	0.03	0.17
HDL															
>42.75 mg/dL	1.159 (0.568–2.365)	0.69	0.69	1.899 (1.010–3.569)	0.05	0.12	1.526 (0.688–3.384)	0.30	0.37	0.383 (0.107–1.368)	0.14	0.23	0.436 (0.218–0.874)	0.02	0.10
≤42.75 mg/dL	0.656 (0.347–1.239)	0.19	0.69	0.907 (0.502–1.641)	0.75	0.88	0.945 (0.443–2.016)	0.88	0.88	1.298 (0.411–4.099)	0.66	0.88	0.691 (0.355–1.343)	0.28	0.69
LDL															
<130 mg/dL	0.908 (0.464–1.779)	0.78	0.78	1.358 (0.714–2.585)	0.35	0.78	1.141 (0.496–2.625)	0.76	0.78	0.805 (0.247–2.630)	0.72	0.78	0.593 (0.293–1.199)	0.15	0.73
≥130 mg/dL	0.238 (0.033–1.721)	0.16	0.39	1.553 (0.273–8.853)	0.62	0.78	4.339 (0.662–28.419)	0.13	0.39	3.547 (0.228–55.280)	0.37	0.61	1.227 (0.201–7.503)	0.83	0.83

Abbreviations: AOR, adjusted odds ratio (adjusted by age, gender, HTN, DM, smoking, folate, homocysteine, BMI, VB_12_, triglycerides, cholesterol, HDL, LDL); CI, confidence interval; HTN, hypertension; DM, diabetes mellitus; BMI, body mass index; VB_12_, vitamin B_12_; HDL, high-density lipoprotein; LDL, low-density lipoprotein. ^a^
*p*-value calculated by multiple logistic regression analysis. ^b^ False-positive discovery rate (FDR)-adjusted *p*-value.

**Table 4 ijms-21-04334-t004:** Haplotype analysis of the *PAI-1* gene polymorphisms on colorectal cancer risk.

Genotype	Controls (2*n* = 832)	Cases (2*n* = 918)	AOR (95% CI)	*p* ^a^	*p* ^b^
*PAI-1* -844G>A/-675 4G > 5G/+43G > A/+9785G > A/+11053T > G					
G-4G-G-G-T	84 (10.1)	99 (10.8)	1.000 (reference)		
G-4G-G-G-G	68 (8.2)	81 (8.8)	1.011 (0.655–1.560)	1.00	1.00
G-4G-G-A-T	6 (0.7)	3 (0.3)	0.424 (0.103–1.749)	0.31	0.59
G-4G-G-A-G	1 (0.1)	1 (0.1)	0.849 (0.052–13.78)	1.00	1.00
G-4G-A-G-T	17 (2.0)	10 (1.1)	0.499 (0.217–1.149)	0.10	0.48
G-4G-A-G-G	9 (1.1)	0 (0.0)	0.045 (0.003–0.780)	0.01	0.01
G-4G-A-A-G	0 (0.0)	3 (0.3)	5.945 (0.303–116.8)	0.25	0.53
G-5G-G-G-T	190 (22.8)	224 (24.4)	1.000 (0.706–1.418)	1.00	1.00
G-5G-G-G-G	44 (5.3)	36 (3.9)	0.694 (0.410–1.177)	0.18	0.53
G-5G-G-A-T	19 (2.3)	26 (2.8)	1.161 (0.601–2.245)	0.74	0.94
G-5G-G-A-G	3 (0.4)	1 (0.1)	0.283 (0.029–2.772)	0.34	0.60
G-5G-A-G-T	30 (3.6)	54 (5.9)	1.527 (0.897–2.602)	0.14	0.53
G-5G-A-G-G	0 (0.0)	1 (0.1)	2.548 (0.102–63.42)	1.00	1.00
G-5G-A-A-T	1 (0.1)	0 (0.0)	0.283 (0.011–7.046)	0.46	0.72
A-4G-G-G-T	61 (7.3)	67 (7.3)	0.932 (0.593–1.466)	0.82	0.99
A-4G-G-G-G	261 (31.4)	271 (29.5)	0.881 (0.629–1.234)	0.49	0.72
A-4G-G-A-T	3 (0.4)	2 (0.2)	0.566 (0.092–3.467)	0.66	0.90
A-4G-G-A-G	0 (0.0)	3 (0.3)	5.945 (0.303–116.8)	0.25	0.53
A-4G-A-G-T	0 (0.0)	5 (0.5)	9.342 (0.509–171.5)	0.07	0.38
A-4G-A-G-G	28 (3.4)	4 (0.4)	0.121 (0.041–0.360)	0.01	0.01
A-4G-A-A-T	2 (0.2)	0 (0.0)	0.170 (0.008–3.590)	0.22	0.53
A-5G-G-G-T	5 (0.6)	22 (2.4)	3.733 (1.354–10.29)	0.01	0.05
A-5G-G-G-G	0 (0.0)	3 (0.3)	5.945 (0.303–116.8)	0.25	0.53
A-5G-A-G-G	0 (0.0)	2 (0.2)	4.246 (0.201–89.74)	0.50	0.72

AOR, adjusted odds ratio (adjusted by age, gender, HTN, DM, smoking, folate, homocysteine, BMI, VB_12_, triglycerides, cholesterol, HDL, LDL). ^a^
*p*-value calculated by Fisher’s exact test. ^b^ False positive discovery rate (FDR) *p*-value.

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
