# Peer review of "Association between Five Common Plasminogen Activator Inhibitor-1 (PAI-1) Gene Polymorphisms and Colorectal Cancer Susceptibility"

_ijms, 2020, doi:10.3390/ijms21124334_

Round 1

Reviewer 1 Report

In this research study, the authors investigated the effect of five PAI-1 gene polymorphisms on the risk of CRC in a Korean population-based case-control study. 

Experimental plan is well designed and performed.

Methods are clear.

Results corroborates the discussion.

Authors also provided limitations to their findings.

Author Response

We thank you for your careful reading of the manuscript.

Kind regards,      

Nam Keun Kim, PhD

Department of Biomedical Science, College of Life Science, CHA University

335 Pangyo-ro, Bundang-gu, Seongnam 13488, South Korea

Tel: +82-31-881-7137, Fax: +82-31-881-7249

Reviewer 2 Report

The study entitled “Association between five common plasminogen activator inhibitor-1 (PAI-1) gene polymorphisms and colorectal cancer susceptibility” analyzed the association between five PAI-1 gene polymorphisms and colorectal cancer (CRC). Overall, the study is quite accurate. Methods fit with the study purpose. Results are clearly presented.

I have no comments.

Author Response

(The authors gave the same response as above.)

Reviewer 3 Report

This manuscript reports the association between polymorphisms in the Plasminogen Activator Inhibitor gene and colorectal cancer.  The authors analyze the polymporphisms' associations with CRC risk and prognosis, as well as their interactions with a variety of clinical factors.  The  work appears to have been done carefully, the analysis is strong, and the manuscript is reasonably well written.  The results will be interesting to the field.

General comments: 

Table 4 should be a supplemental table.  The important results in this table are conveyed well by Figure 1.

The discussion of linkage disequilibrium in lines 151-155 is a surprise, as this is the first mention of linkage disequilibrium.  Supplemental Figure 1 should be included as a regular figure and described in the Results section.  (Alternatively, it could be deleted from the supplemental figures and the linkage disequilibrium discussion removed from the Discussion.  It is interesting but not essential.)

Specific comments:

Generally, numbers should be given to no more than two significant digits (e.g. P = 0.25, not 0.252).  The additional digit is not justified.

Line 36:  Recommend deleting "followed by thyroid cancer"

Line 41:  Fibrinolysis should be defined

Lines 45/46:  The role of PAI in the inhibition of tissue and urinary plasminogen activators is repeated almost verbatim in Line 48.  One of these should be deleted.  In addition, the relationship between fibrinolysis and plasminogen activators should be described (e.g. that plasmin cleaves fibrin and other proteins involved in blood vessel maintenance)

Line 79:  "advanced stage disease stage" – the second "stage" should be deleted

Table 1:  What does "male/female" in "HDL-C <40/50 (male/female, mg/dl) mean?"  If it means the data reflect both males and females, it would be better to delete it (i.e. HDL-C <40/50 (mg/dl))

Table 2:  Define "HWE."  This could be done in the text in line 83  "Hardy Weinberg equilibrium (HWE; p>0.05)"

Line 85:  Explain that Supplemental Table 1 analyzes colon and rectum data separately.

Line 85:  Recommend deleting "As a result of the CRC SNPs, we hypothesized that 5G5G increases CRC risk."  This is stated more appropriately in Lines 89-90.

Line 100-103:  Recommend stating the remaining significant results:  The association of 4G5G+5G5G with higher CRC risk in patients with low triglycerides and high HDL; the association of 11053TG+GG with protection against CRC in patients with high cholesterol and low HDL; and the association of 9785 GA with elevated risk for those >61 years old.

Line 105:  The reference to Figure 1 and Table 4 should be moved from the end of Line 107 to the end of this line (e.g. "with a folate level ≤3.8 nmol/L (Figure 1 and Supplemental Table X).")

Lines 149-152:  The meaning here is unclear; please clarify.

Line 172:  "The mechanism of PAI-1 gene involvement in polymorphism in cancer spread" is unclear; please clarify.

Line 181:  Recommend replacing "despite having the same" with "independent of"

Line 182:  Recommend deleting "Although they have the same polymorphism"

Line 191:  Please add a sentence discussing how the influence of clinicopathological features on CRC, and their interactions with specific polymorphisms, might explain variable and apparently contradictory results in other studies (refs 32-27).

Line 193:  The meaning of this sentence is unclear; please clarify.

Line 282:  Recommend separating this "Conclusions" paragraph from the Materials and Methods.

Author Response

We thank you for your careful reading of the manuscript and helpful comments and suggestions. We have made revisions according to your comments and suggestions.

Enclosed you will find the revised version of our manuscript “Association between five common plasminogen activator inhibitor-1 (PAI-1) gene polymorphisms and colorectal cancer susceptibility” submitted for publication in IJMS. The revised parts have been indicated in blue font in the revised manuscript.

General comments: 

â–  Table 4 should be a supplemental table.  The important results in this table are conveyed well by Figure 1.

Thank you for kind comment. According to your suggestion, we moved table 4 to the supplement table. Please see supplement table 5.

â–  The discussion of linkage disequilibrium in lines 151-155 is a surprise, as this is the first mention of linkage disequilibrium.  Supplemental Figure 1 should be included as a regular figure and described in the Results section.  (Alternatively, it could be deleted from the supplemental figures and the linkage disequilibrium discussion removed from the Discussion.  It is interesting but not essential.)

→ Thank you for kind comment. According to your suggestion, we deleted the sentences in the Discussion as follows:

In our results, there were linkage disequilibrium (LD) patterns between the five SNPs selected for haplotype analysis, and the allele combination analysis showed that the CRC risk increased in some cases, even though the combination of minor alleles was shown to reduce this risk. This result may have been caused by classifying haplotypes using a small sample size. After performing statistical analyses on these results, it will be necessary to repeat the study with a larger patient population Supplementary Figure 1.

Specific comments:

  • Generally, numbers should be given to no more than two significant digits (e.g. P = 0.25, not 0.252).  The additional digit is not justified.

→Thank you for kind comments. We revise the Table1-4 as follows:

We Change to 2 digits of P-value significant digits.

please see Table1-4 ,Supplementary Table 1-4 section.

â–  Line 36:  Recommend deleting "followed by thyroid cancer"

 → Thank you for kind comment. According to your suggestion, we deleted "followed by thyroid cancer" in the Introduction.

  • Line 41:  Fibrinolysis should be defined

→ Thank you for critical comments. We added the sentence in the introduction as follows:

Fibrinolysis is a normal body process. It prevents blood clots that occur naturally from growing and causing problems (4)

<Reference>

  1. Lijnen HR, Collen D. Mechanisms of physiological fibrinolysis. Baillieres Clin Haematol. (1995) 8, 277-290. 10.1016/s0950-3536(05)80268-9 (1995).

  • Lines 45/46:  The role of PAI in the inhibition of tissue and urinary plasminogen activators is repeated almost verbatim in Line 48.  One of these should be deleted.  In addition, the relationship between fibrinolysis and plasminogen activators should be described (e.g. that plasmin cleaves fibrin and other proteins involved in blood vessel maintenance)

→ Thanks for your kind comments. According to your suggestion, the sentence in introduction line 48 has been deleted.

We added sentences in the introduction as follows:

The regulation of the fibrinolytic system is of critical importance during hemostasis, wound repair, neoplasia, inflammation, and a variety of other biologic processes. This control is achieved in a large part through the action of specific plasminogen activator inhibitors (PAIs). Cultured endothelial cells (ECs) produce type 1 PAI (PAI-1), the physiologic inhibitor of tissue-type plasminogen activator. PAI-1 is one of the most highly regulated of the fibrinolytic components produced by ECs. Its synthesis is modulated by a variety of compounds including endotoxin, thrombin, transforming growth factor beta interleukin 1, and tumor necrosis factor alpha. (12)

<Reference>

  1. Schleef RR, Loskutoff DJ. Fibrinolytic system of vascular endothelial cells. Role of plasminogen activator inhibitors. Haemostasis. (1988) 18, 328-341. 10.1159/000215815

  • Line 79:  "advanced stage disease stage" – the second "stage" should be deleted

→Thank you for kind comment. We deleted the second "stage". Please see Results section.

  • Table 1:  What does "male/female" in "HDL-C <40/50 (male/female, mg/dl) mean?"  If it means the data reflect both males and females, it would be better to delete it (i.e. HDL-C <40/50 (mg/dl))

→ Thank you for kind comment. According to your suggestion, the sentence of HDL-C <40/50 (male/female, mg/dl) Table 1 ‘Results’ has been deleted.

  • Table 2:  Define "HWE."  This could be done in the text in line 83  "Hardy Weinberg equilibrium (HWE; p>0.05)"

→Thank you for kind comments. We revise the sentences as follows:

The genotype and allele frequencies were in Hardy-Weinberg equilibrium (p>0.05) in the control and patients individuals for all polymorphisms (Table 2).

  • Line 85:  Explain that Supplemental Table 1 analyzes colon and rectum data separately.

→Thank you for kind comments. We added sentences in the Results as follows:

We found a lower risk of PAI-1 +11053T>G with the GG genotype (AOR=0.594; 95% CI: 0.370 - 0.954; p=0.031) than with the TT genotype, as well as for recessive carriers (TT+TG vs. GG, AOR=0.667; 95% CI: 0.446-0.990; p=0.049) (Supplemental Table 1).

  • Line 85:  Recommend deleting "As a result of the CRC SNPs, we hypothesized that 5G5G increases CRC risk."  This is stated more appropriately in Lines 89-90.

→Thank you for kind comments. We deleted the sentences in the Results as follows:

"As a result of the CRC SNPs, we hypothesized that 5G5G increases CRC risk."

  • Line 100-103:  Recommend stating the remaining significant results:  The association of 4G5G+5G5G with higher CRC risk in patients with low triglycerides and high HDL; the association of 11053TG+GG with protection against CRC in patients with high cholesterol and low HDL; and the association of 9785 GA with elevated risk for those >61 years old.

→Thank you for kind comments. We added the sentences in the Results as follows: “However, after considering multiple comparisons, we concluded that The association of 4G5G+5G5G, the association of 11053TG+GG and the association of 9785 GA polymorphism had no relation with CRC”

  • Line 105:  The reference to Figure 1 and Table 4 should be moved from the end of Line 107 to the end of this line (e.g. "with a folate level ≤3.8 nmol/L (Figure 1 and Supplemental Table X).")

→Thank you for kind comments. We moved the Figure 1 and Supplemental Table 5 to the end of line "with a folate level ≤3.8 nmol/L”. please see Results section.

  • Lines 149-152:  The meaning here is unclear; please clarify.

→ Thank you for kind comments.

This is just an interpretation of table 4 and 2.4. ‘The five combined PAI-1 polymorphisms have different effects on CRC occurrence’ in result section.

According to your suggestion, we deleted the sentences in the Discussion as follows:

“In our results, there were linkage disequilibrium (LD) patterns between the five SNPs selected for haplotype analysis, and the allele combination analysis showed that the CRC risk increased in some cases, even though the combination of minor alleles was shown to reduce this risk. This result may have been caused by classifying haplotypes using a small sample size. After performing statistical analyses on these results, it will be necessary to repeat the study with a larger patient population Supplementary Figure 1”.

  • Line 172:  "The mechanism of PAI-1 gene involvement in polymorphism in cancer spread" is unclear; please clarify.

→Thank you for kind comments. We revised the sentence as follows:

"The mechanism of PAI-1 gene involvement in polymorphism in cancer spread pathogenesis may directly result from opposing effects of the two alleles at the transcription level"

  • "The mechanism of how polymorphisms of PAI-1 gene affects the progression and metastasis of cancer may directly result from opposing effects of the two alleles at the transcription level" (line 176)

  • Line 181:  Recommend replacing "despite having the same" with "independent of"

→Thank you for kind comments. We revised the sentence as follows:

"independent of".

  • Line 182:  Recommend deleting "Although they have the same polymorphism"

→Thank you for kind comments.

We deleted "Although they have the same polymorphism":

Please see discussion.

  • Line 191:  Please add a sentence discussing how the influence of clinicopathological features on CRC, and their interactions with specific polymorphisms, might explain variable and apparently contradictory results in other studies (refs 32-27).

→Thank you for kind comments. We added the sentences in the as follows:

"The variable and contradictory results of other studies [32-37] may be another evidence that interactions with other factors, such as environmental factors as well as genes, in the development of cancer have an influence. However further specifically designed studies are needed to assess its value in this respect." (line 171-174)

  • Line 193:  The meaning of this sentence is unclear; please clarify.

→Thank you for kind comments. We revised the sentence as follows:

"Thus far, the association between PAI-1 3′-UTR polymorphisms of including CRC has not been investigated. Our association study of PAI-1 3′-UTR polymorphism (rs7242) is the first report of CRC." à

"The rs7242 polymorphism is located in the 3′-untranslated region (UTR) of PAI-1 and is characterized by the substitution of a guanine with thymine. Our association study of PAI-1 3′-UTR polymorphism (+11053T>G) is the first report of CRC." (line 198-200)

  • Line 282:  Recommend separating this "Conclusions" paragraph from the Materials and Methods.

→Thank you for kind comments. We deleted "conclusions" from the Materials and Methods and we moved the conclusions to the end of Discussion. Please see discussion.